# Bone Pain in Cancer Patients: Mechanisms and Current Treatment

**DOI:** 10.3390/ijms20236047

**Published:** 2019-11-30

**Authors:** Renata Zajączkowska, Magdalena Kocot-Kępska, Wojciech Leppert, Jerzy Wordliczek

**Affiliations:** 1Department of Interdisciplinary Intensive Care, Jagiellonian University Medical College, 31-008 Krakow, Poland; j.wordliczek@uj.edu.pl; 2Department of Pain Research and Treatment, Jagiellonian University Medical College, 31-008 Krakow, Poland; 3Laboratory of Quality of Life Research, Chair and Department of Palliative Medicine, Poznan University of Medical Sciences, 61-701 Poznan, Poland; wojciechleppert@wp.pl

**Keywords:** cancer-induced bone pain, nociceptive pain, neuropathic pain, multimodal pain treatment

## Abstract

The skeletal system is the third most common site for cancer metastases, surpassed only by the lungs and liver. Many tumors, especially those of the breast, prostate, lungs, and kidneys, have a strong predilection to metastasize to bone, which causes pain, hypercalcemia, pathological skeletal fractures, compression of the spinal cord or other nervous structures, decreased mobility, and increased mortality. Metastatic cancer-induced bone pain (CIBP) is a type of chronic pain with unique and complex pathophysiology characterized by nociceptive and neuropathic components. Its treatment should be multimodal (pharmacological and non-pharmacological), including causal anticancer and symptomatic analgesic treatment to improve quality of life (QoL). The aim of this paper is to discuss the mechanisms involved in the occurrence and persistence of cancer-associated bone pain and to review the treatment methods recommended by experts in clinical practice. The final part of the paper reviews experimental therapeutic methods that are currently being studied and that may improve the efficacy of bone pain treatment in cancer patients in the future.

## 1. Introduction

Bone pain is one of the most common types of pain in cancer patients [1]. Approximately 60–84% of patients with advanced cancer are estimated to experience varying degrees of bone pain [2]. This condition affects millions of patients worldwide, with nearly 450,000 patients annually in the USA alone [3].

Bones are the third most frequent (after the lungs and liver) target sites of metastases [4]. The most common bone metastases arise from multiple myeloma, as well as cancer of the breast, prostate, lungs, thyroid, kidneys, and ovaries [5]. It is estimated that pathological changes in the bones occur in 70% of patients at the time of the diagnosis of the disease and in 90% of all patients during the course of multiple myeloma [6]. Cancer of the breast, lungs, and prostate are jointly responsible for 80% of cancer metastases to the bones [7]. As many as 65% of all bone metastases originate from cancer of the breast in women and from cancer of the prostate in men. The remaining 35% of metastasis cases arise from cancer of the kidneys, thyroid, and lungs [8]. The relative incidence of bone metastases is 65–75% in breast cancer, 65–75% in prostate cancer, 60% in thyroid cancer, 40% in bladder cancer, 20–25% in renal cell carcinoma, and 14–45% in melanoma [9].

Cancer metastases to the skeletal system are most often located in the vertebrae (69%), followed by the pelvic bones (41%), long bones (usually the proximal femur) (25%), and skull (14%). They occur less frequently in the ribs, sternum, and proximal humerus [10]. Most often, dissemination takes place through the bloodstream; it occurs less often through infiltration from surrounding tissues or through the lymphatic system or cerebrospinal fluid (the latter path affects children more often) [11].

It should be emphasized that the location and severity of metastatic bone lesions do not always correlate with the severity of pain experienced by cancer patients. Some patients presenting with disseminated bone lesions experience low to moderate pain, whereas others with a single lesion report severe or very severe pain [12]. These observations warrant an individualized approach to each patient treated for metastatic bone pain.

## 2. Clinical Characteristics of Bone Pain in Cancer Patients

In approximately 20% of patients, cancer initially develops without symptoms, and pain or a pathological bone fracture constitutes the first symptom of the disease [13]. Sometimes, pain precedes the onset of radiographically detectable changes in bones. Usually, pain occurs spontaneously, and it varies in severity and character depending on the disease stage. Most patients initially experience intermittent dull aches, but as the disease progresses, pain becomes constant and more severe. Its intensity cannot be predicted by the tumor type, the tumor size, the number of metastases, or bone involvement [12,14]. Bone pain intensifies during movement and can be accompanied by fever. Typically, pain also increases in severity at night [13]. Pain upon palpation is often found in the area of metastatic bone lesions.

Continued tumor growth within the bone usually leads to another type of cancer pain: breakthrough (episodic) pain. Breakthrough (episodic) pain is defined as recurrent episodes of extreme pain breaking through the regimen administered to treat background pain [2]. Its clinical manifestation comprises a temporary intensification of pain experienced by patients with stable and effectively treated background pain [5]. It is usually acute, piercing, and very severe. Breakthrough (episodic) pain can be spontaneous—it may occur without obvious triggers—or incidental, induced by various factors (commonly by movement and body weight-bearing) [15]. In everyday clinical practice, observations of patients with bone metastases reveal that breakthrough (episodic) pain often poses a greater therapeutic problem than background pain. This is caused not only by epidemiological reasons—it is estimated that breakthrough (episodic) pain occurs in approximately 75% of patients with bone metastases—but also by the temporal aspect of this type of pain (in more than half of patients, the time to maximum pain intensity is very short—less than 5 minutes—and the pain lasts less than 15 minutes), as well as its unpredictability, severe intensity, and negative impact on daily functioning and quality of life (QoL) [16]. These factors play a crucial role in the selection of drugs to treat breakthrough (episodic) pain.

At a later stage, when the destruction of bone is extensive, pathological fractures with concomitant compression and damage to nervous system structures (spinal cord, nerve roots, plexuses, or peripheral nerves) may occur. The latter mechanism is one of the causes of the neuropathic component of bone pain in cancer patients [17].

## 3. Mechanisms of Bone Pain in Cancer Patients

The mechanism of pain experienced by patients as a result of metastases to the bone is complex. It involves various interactions between tumor cells, bone cells, activated inflammatory cells, and bone-innervating neurons. It includes inflammatory and neuropathic processes, which are modified at the level of peripheral tissues and nerves, as well as at higher levels of the nervous system (the spinal cord and brain).

Interestingly, the incidence and severity of bone pain are not always proportional to the number and size of bone metastases. Approximately 25% of patients with such metastases feel no pain. Some lesions cause mild complaints, whereas others, although single or small in size, cause very severe pain in the absence of pathological fractures [18,19]. This diversity of symptoms is the result of the different contributions of central and peripheral mechanisms and different possibilities to reduce peripheral activity by central modulation systems, which show significant individual differences [20].

The first stage of bone metastases involves the separation of cancer cells from the primary tumor mass and their penetration into the systemic circulation. Tumor cells undergo epithelial–mesenchymal transition (EMT). That is, their phenotype changes from epithelial to mesenchymal as a result of the loss of intercellular adhesion proteins on cell surfaces [21]. An important part of this process is played by the matrix metalloproteinases (MMPs) produced by tumor cells. MMPs contribute to the degradation of the extracellular matrix, and thus, they facilitate the escape of tumor cells from the primary tumor mass through the surrounding tissues into the lumen of neighboring blood vessels [22]. Besides the elevated levels of matrix metalloproteinases (which remodel the tumor extracellular matrix), tumor cells and activated fibroblasts secrete vascular endothelial growth factor (VEGF) and CXCL-family chemokines (e.g., CXCL12 and CCL2), which result in the recruitment of leukocytes and endothelial cells into the tumor microenvironment [3]. Recruited vascular cells form an aberrant tumor vasculature characterized by increased permeability and expression of cell adhesion molecules (CAMs), which promote the intravasation of tumor cells [23].

After entering the systemic circulation, tumor cells face the challenge of surviving in the new, alien, and hostile environment of the vascular bed [24]. They defend themselves by inhibiting the physiological process of programmed cell death, which cancer cells undergo if they lose cell–matrix or cell–cell interactions [25]. This inhibition occurs by means of overexpressing tropomyosin receptor kinase B (TrkB) on the tumor cell membrane, which results in the activation of phosphatidylinositol-4,5-bisphosphate 3 kinase (PI3K)–AKT pro-survival pathways [26]. Another mechanism that prevents the destruction of tumor cells by macrophages involves upregulating the expression of certain cell-surface proteins, such as cluster of differentiation (CD) 47 [27].

The mechanisms responsible for “homing” the circulating tumor cells to bone are complex and incompletely understood. An important part of this phenomenon is played by the system CXCL12–CXC–chemokine receptor 4 [28]. CXCL12, also called stromal-derived factor–1 (SDF-1), is a chemokine factor that activates several divergent intracellular signaling pathways involved in a variety of cellular processes (cell survival, gene transcription, chemotaxis, and integrin expression, such as integrin αVβ3 on the surface of circulating tumor cells) [28]. It has been shown that increased expression of αVβ3 on the surface of metastatic prostate tumor cells promotes the adherence of tumor cells to endothelial cells in the bone marrow [29]. Although the precise mechanisms are unclear, it appears that upon arrival in bone, some cancer cells enter a state of dormancy, which can apparently last for many months if not years.

Once settled in bone, tumor cells divide, and the growing tumor mass causes progressive bone damage. Tumor cells, stromal cells, and inflammatory cells recruited by tumor cells (macrophages, neutrophils, T cells, mast cells) produce and release several mediators, and osteoblasts, osteoclasts, and nerve fibers that supply the bone serve as effectors [30]. These mediators include endothelin, bradykinin, proteases, interleukin (IL) 6, hydrogen ions (H^+^), colony-stimulating factors (CSFs), nerve growth factor (NGF), prostaglandin (PG) E2, serotonin, and tumor necrosis factor (TNF) α [30]. They can sensitize or activate bone-innervating sensory nerve endings. Information on the pathological process and the resulting damage is passed on to the spinal cord and then to the brain, where perception takes place [1].

The adult bone receives restricted and unique innervation. All the bone compartments (marrow, mineralized bone, and the overlying periosteum) are innervated by sensory nerve fibers, which differ significantly in terms of their density. The periosteum is the most densely innervated, whereas the fewest nerve fibers are in mineralized bone. For every 100 nerve fibers in the periosteum, 2 nerve fibers can be found in the bone marrow and 0.1 in mineralized bone [14]. Sensory nerves that innervate bones are predominantly thinly myelinated and unmyelinated nerve fibers: A-δ (delta) or C-sensory nerve fibers (80% tropomyosin kinase A (TrkA)-positive) [14]. There are several types of receptors and ion channels on the sensory nerves that supply bones, and they help nociceptors to detect and transmit signals from the noxious stimuli produced by cancer and tumor-associated immune cells. Endothelin receptor (ET_A_R), prostaglandin (PG) receptors, TrkA receptors, bradykinin receptors, cytokine receptors, chemokine receptors, TRPV1 (transient receptor potential channel, vanilloid subfamily member 1), acid-sensing ion channel 3 (ASIC3), and purinergic receptor (P2X3) are activated by tumor cells, stromal cells, and activated immune cell mediators, including endothelin (ET), PGE2, NGF, bradykinin, and pro-inflammatory cytokines (TNFα, interleukins (IL1β, IL6, IL8, IL15), chemokines (CCL5, monocyte chemoattractant protein-1 (MCP-1), macrophage inflammatory protein-1a (MIP-1a)), H^+^, and extracellular adenosine triphosphate (ATP) (Figure 1) [14].

The processes that occur at the site of bone metastases result from the interactions between the tumor cells attacking the bone, osteoclasts, osteoblasts, and inflammatory cells [31,32]. Tumor cells release endothelin (ET), which, through its appropriate receptors, interacts with osteoblasts to stimulate their proliferation. This leads to the formation of new bone and growth, but such bone is weak and prone to fracture. Activated osteoblasts release receptor activator of nuclear factor kappa B ligand (RANKL), which, in turn, signals the proliferation and maturation of osteoclasts [33]. Osteoclasts produce ATP and acidosis-causing H^+^, which activate the appropriate receptors (P2X, TRPV1, ASICS3) located on bone-supplying neurons [12]. Osteoclasts also secrete collagenases and proteases, thereby contributing to bone demineralization and damage to bone matrix protein, as well as growth factors, such as transforming growth factor-beta (TGFβ) and insulin-like growth factor (IGF), which promote the proliferation of tumor cells by inhibiting their apoptosis [33,34]. Acidosis, which is caused mainly by tumor cells and osteoclasts, activates the above-mentioned channels (TRPV1, ASICS3) and stimulates stromal cells to produce and release not only growth factors (e.g., NGF, brain-derived neurotrophic factor (BDNF)) but also pro-inflammatory mediators (e.g., IL1β, IL6, IL15, CCL5) [12]. NGF and BDNF are also released by tumor cells. They stimulate macrophages to produce pro-inflammatory cytokines (TNFα, IL1β, IL6) and prostaglandins, which induce pain by binding to their receptors on sensory neurons [14]. The interactions between tumor cells, osteoblasts, osteoclasts, stromal cells, and inflammatory cells are shown in Figure 2.

In health, physiological bone remodeling is reliant on the balance between bone-forming osteoblast activity and bone resorption by osteoclasts [5]. This balance is disturbed in neoplastic bone transformation [13,24]. At radiological and histopathological levels, bone metastases can be classified as osteolytic, osteosclerotic, or mixed [35]. Typically, osteolytic lesions are caused by breast cancer and multiple myeloma. These lesions can be particularly dangerous and are associated with the highest risk of fracture [25]. Osteoblastic lesions most commonly occur secondary to bone metastases caused by cancer of the prostate. Osteoblastic bone remodeling increases the risk of pathological fractures but to a lesser extent than that observed for osteolytic metastases [25]. Rather than strictly resorbing or forming bone, metastatic lesions appear to stimulate osteoblastic and osteolytic activity to varying degrees [13]. However, regardless of the type of metastases (osteolytic, osteosclerotic, or mixed), progressive damage to normal bone structure, impaired bone stability, and mechanical resistance may lead to pathological fractures and their attendant consequences.

The process of bone metastasis is crucially dependent on communication between bone-attacking tumor cells, bone matrix cells, and bone-innervating nerve fibers. Tumor cells do not damage bone tissue directly; instead, they primarily activate the RANKL/RANK (receptor activator for nuclear factor kappa B) system by producing receptor activator for RANKL, which, having bound with RANK on the surface of osteoclasts, initiates their proliferation and, in consequence, triggers their damaging effect on bones [1]. RANKL is a protein member of the cytokine family of tumor necrosis factors produced by tumor cells, the osteoblastic cell line (i.e., by mature osteoblasts and their precursors), and activated T cells. RANKL expression is dependent upon a host of factors, including cytokines (IL1, IL6, IL11, TNFα) and glucocorticosteroids. The binding of RANKL to RANK likely plays a key part in the proliferation, differentiation, and maturation of osteoclasts, which resorb bone by forming a highly acidic resorption ‘bay’ or ‘pit’ between the osteoclast and bone. Resorption-induced acidosis stimulates the channels TRPV1 or ASIC3 and triggers cancer-induced bone pain (CIBP) [36].

The discovery of this mechanism has clinical implications. Recent years have seen the introduction of denosumab, a human monoclonal immunoglobulin G2 antibody against RANKL, into clinical practice. Because of its high affinity and specificity, denosumab binds to RANKL and prevents the activation of the RANK receptor on the surface of osteoclasts and their precursors, thereby inhibiting their formation, proliferation, and survival and reducing osteoclastic bone resorption. Numerous studies conducted in recent years have supported the efficacy of denosumab in relieving pain due to bone metastases, reducing the incidence of pathological fractures, and improving the quality of life (QoL) and daily functioning of patients with bone metastases (Figure 3) [37,38].

Local acidosis plays an important part in tumor-induced bone destruction and CIBP [12]. It is triggered mainly by osteoclasts (which dissolve bone-building minerals and damage the organic matrix) and tumor cells. Osteoclasts resorb bone through highly acidic ‘bays’ between osteoclasts and bone. These ‘bays’ release protons, which stimulate acid-sensitive channels, such as TRPV1 and ASIC, expressed by sensory bone-innervating neurons. Tumor cells undergoing the Warburg mechanism, which protects them from intracellular acidosis, release H^+^ and lactates outside the cell. H^+^ from the sources described above stimulates bone-innervating sensory neurons by activating acid-sensitive ionic channels, primarily TRPV1 and ASIC3 [12]. Information on their activation is then transmitted to the upper levels of the nervous system through the spinal cord to the brain, where pain is perceived.

The invasion of bone by tumor cells initiates the pathological growth and formation of a network of new nerve fibers with a unique morphology, organization, and density, which exceeds the density of nerve fibers in normal bone by up to 10–70 times [39,40]. These new pathological nerve fibers are formed within the periosteum, mineralized bone tissue, and the bone marrow as a result of the activity of mediators produced and released by tumor cells. It has been shown that NGF plays a major part in the creation of such pathological networks [39,40]. NGF belongs to a family of neurotrophins. It is synthesized in tissues surrounding the nerves and ensures appropriate nerve growth. In health, NGF-synthesizing cells include eosinophils, mast cells, macrophages, keratinocytes, and Schwann cells [41].

Under pathological conditions, NGF is produced by tumor, inflammatory, and immune system cells. NGF affects its target cells by binding to two types of membrane receptors: TrkA (exclusively specific for NGF) and p75NTR (a receptor common to all neurotrophins) [42]. When bound to the TrkA receptor, NGF modulates the release of neurotransmitters (e.g., substance P, calcitonin gene-related peptide (CGRP)), activates ion channels (TRPV1, ASIC–3, P2X3, sodium channels) and receptors (e.g., for bradykinin), and regulates the synthesis of structural particles (e.g., neurofilaments). NGF is also responsible for the pathological sprouting and reorganization of sensory and sympathetic nerve fibers, as well as the formation of neuroma-like structures within them. The latter mechanism is most likely responsible for breakthrough paroxysmal pain and its exacerbation upon movement or activity in patients with bone metastases [42]. This sprouting appears to require NGF because sustained administration of anti-NGF or a pan-Trk inhibitor largely blocks the pathological sprouting of sensory nerve fibers and the formation of neuroma-like structures and significantly inhibits pain generation [14,43].

In normal bone, sensory and sympathetic nerve fibers are separate from each other. The pathological tumor-induced growth of sensory and sympathetic nerve fibers observed in bone metastases leads to their reorganization and the formation of connections between these nerve fiber types; as a result, pain stimuli can also be caused by the activation of sympathetic fibers [39].

The discovery of these mechanisms has enabled researchers to search for new treatment methods of bone pain in cancer patients. Tanezumab is a humanized monoclonal anti-NGF antibody that prevents NGF from activating TrkA receptors. Consequently, it inhibits the pathological sprouting and formation of new pathological nerve fiber networks and thus may relieve pain in patients with metastatic bone pain [20,30,37,43] (Figure 4).

## 4. Treatment of Cancer-Induced Bone Pain

Cancer-induced bone pain (CIBP) is multifactorial and involves several mechanisms, so different strategies for different disease stages and pain mechanisms should be applied to achieve optimal pain relief and QoL. The therapeutic goal is not only pain relief but also the prevention of pain progression and skeletal-related events (SREs). Therefore, treatment of bone pain in cancer patients should be multimodal (pharmacological and non-pharmacological), including causal anticancer and symptomatic analgesic treatment [44,45]:✓Anticancer treatment comprises local surgery and/or radiotherapy (RT) and systemic treatment modalities (chemotherapy, hormone therapy, immunotherapy, and molecular treatment). Causal treatment reduces the tumor mass and local tissue infiltration and thus decreases the pain intensity; hence, anticancer treatment may be regarded as symptomatic treatment, as well.✓Analgesic treatment includes pain management with analgesics (non-opioid and opioid analgesics according to the World Health Organization (WHO) analgesic ladder), bone-targeted therapies (NGF inhibitors; osteoclast inhibitors, such as bisphosphonates and denosumab), and adjuvants (corticosteroids, anticonvulsants).

All interventions should be individualized and directed at relieving pain, improving QoL, increasing the functioning of patients, and, whenever possible, prolonging survival. The therapeutic options for pain due to bone metastases according to the European Society for Medical Oncology (ESMO) and the World Health Organization (WHO) are presented in Figure 5.

### 4.1. Radiotherapy

According to WHO recommendations, radiotherapy (RT) is used to reduce analgesic requirements, improve quality of life (QoL), as well as maintain or improve skeletal function by reducing the risk of pathological fractures or metastatic spinal cord compression (mSCC). There is a strong recommendation based on high-quality evidence that single-dose, low-fractionated RT should be used in adults with pain related to bone metastases if indicated and available. All patients with bone metastases should be considered for external beam radiotherapy (EBRT) or radioisotope treatment [45].

Typically, for localized pain, EBRT is used with different dosing schedules. However, in terms of pain relief, as well as the onset and duration of pain relief, the efficacy of a single fraction (low-fractionated) of 8 Gy is equal to that of longer schedules, such as 5 fractions of 4 Gy (20 Gy) or 10 fractions of 3 Gy (30 Gy) (high-fractionated) [45,46]. According to clinical data, under both schedules, approximately 25% of patients achieved total pain relief, and approximately 70% achieved total or partial pain relief. Typically, pain relief is experienced by patients within 1–2 weeks [47]. Expert panels [44] have recommended re-irradiation with a dose of 8 Gy in patients with recurrent bone pain. Low-fractionated RT is also cost-effective for patients and clinics, even if re-irradiation is included. However, low-fractionated RT is associated with a higher incidence of pathological fracture at the site of irradiation (relative risk = 1.48; 95% confidence interval, 1.08–2.03) [45].

EBRT may not be efficacious for patients with widespread pain that is difficult to localize [48]. Systemic administration of radiopharmaceuticals (e.g., strontium-89, samarium-153, rhenium, radium-223) may be offered to patients with diffuse bone pain due to osteoblastic or mixed osteoblastic–osteolytic metastases, which cannot be treated efficaciously with EBRT. Pain relief usually starts 1–4 weeks after treatment initiation. Most studies have been conducted in men diagnosed with prostate cancer and shown some effect on pain relief [49]. Bone marrow toxicity, renal function, leukopenia, and thrombocytopenia should be looked for carefully. RT with radium-223 may be recommended for patients with castration-resistant cancer of the prostate because it reduces SREs, decreases pain, and improves QoL and survival [44,50]. However, the current WHO guidelines make no recommendation for or against the use of radioisotopes, mainly because of their cost and because the evidence originates from studies in prostate cancer only [45].

EBRT is the first-line treatment for most patients with mSCC because it provides pain relief in 50–58% of treated patients. The efficacy of doses of 20 Gy in five fractions or 8 Gy in one or two fractions is equivalent to that of more prolonged schedules. For patients who have a longer predicted life expectancy (more than 6 months), higher dose schedules may be considered [44].

### 4.2. Interventional Methods

According to ESMO guidelines, surgery should be considered for patients with mSCC, particularly those with spinal instability, unknown primary histology, recurrence after previous RT, or a solitary site of compression [44]. The optimal method for the surgical treatment of spinal metastases has not been determined, but posterolateral fusion with autologous bone grafting is preferred [51]. Vertebroplasty or kyphoplasty should be considered for patients who have vertebral metastases and show no evidence of mSCC or spinal instability, as well as patients who present with mechanical pain resistant to analgesia or the collapse of vertebral bodies [52].

Minimally invasive procedures, such as balloon kyphoplasty or percutaneous vertebroplasty, may be beneficial in patients who have vertebral metastases without neurological compromise but with persistent pain. Kyriakou et al. [53] reported beneficial results in the treatment of painful metastatic cancer and myeloma-related vertebral compression fractures. Cement augmentation resulted in rapid and sustained pain relief at 1 year, improved back function, improved QoL, increased activity, reduced use of analgesics, and fewer days of bed rest [53]. Recent developments include the use of minimally invasive thermoablation (radiofrequency ablation plus cryoablation methods). For instance, painful bone metastases can be treated using microwave ablation or magnetic resonance-guided high-intensity focused ultrasound [54].

### 4.3. Bone-Targeting Therapies

#### 4.3.1. Bisphosphonates

Bisphosphonates (BPs) have a direct apoptotic effect on osteoclasts. They inhibit their differentiation and maturation and, thus, prevent bone resorption and hypercalcemia, decrease the incidence of SREs, as well as reduce bone pain and the use of analgesics. Their mechanisms of action are shown in Figure 6.

According to WHO recommendations, there is strong evidence from moderate-quality studies that BPs should be used to prevent and treat bone pain in adults (including the elderly) [45]. BPs should be used with analgesics in patients with a good prognosis, particularly if the pain is not localized or RT is not readily available [44]. Clinical studies have shown significant pain relief and reduced risk of SREs with the use of bisphosphonates, but no significant changes in QoL scores have been found. Data from clinical studies do not support the advantage of one bisphosphonate over another. The pain reduction and duration of pain relief were similar for each of the bisphosphonates studied (clodronate, ibandronate, pamidronate, zoledronate). However, clinicians should take into account variable renal effects when prescribing bisphosphonates. Osteonecrosis of the jaw is another serious adverse effect [45]. Consensus regarding the optimal start and duration of treatment is lacking. According to the American Society of Clinical Oncology, treatment with BPs should be started as soon as bone metastases are diagnosed, and they should be continued until there is a substantial decline in the general status of a patient [55,56].

#### 4.3.2. Monoclonal Antibodies

Denosumab is a human, monoclonal, synthetic antibody that binds to RANKL to prevent its interaction with RANK (Figure 3). The binding of RANKL to RANK is required for physiological and tumor-induced proliferation and maturation of osteoclasts [1,36,57]. Treatment with denosumab reduces osteoclast function and thereby delays SREs and the recurrence of bone pain, improving the QoL and functioning of patients with bone metastases [37,38]. According to ESMO, denosumab is indicated as an alternative to bisphosphonates for patients with bone metastases from solid tumors and myeloma. Dental measures to prevent osteonecrosis of the jaw are required before denosumab administration [44]. Studies comparing bisphosphonates with denosumab have suggested that denosumab reduces the risk of SREs and improves functional outcomes more than bisphosphonates, but it increases the risk of osteonecrosis of the jaw and does not influence bone pain or time to pain relief [38,45].

Tanezumab is a recombinant humanized monoclonal antibody. It binds circulating and local tissue NGF to prevent its interaction with TrkA receptors and p75 receptors (Figure 4). The interaction of NGF with TrkA receptors is crucial in nociception because it modulates the expression and function of the sodium channels Nav 1.8 and TRPV1 and increases the synthesis of pronociceptive substances (substance P, CGRP) and sodium channel proteins, which can ultimately result in hyperalgesia [14,20,37,40,41,42,58,59]. In bone cancer, several tumor-associated stromal cells (macrophages, T lymphocytes, mast cells, endothelial cells) can express and release NGF [14,20,59]. Preclinical studies have revealed that therapies that block NGF or TrkA, when given after tumor-induced sprouting and/or neuroma formation, can inhibit pathological sprouting and/or neuroma formation [43]. Tanezumab has been studied extensively in patients with musculoskeletal chronic non-cancer pain, but data on its efficacy in cancer-induced pain are sparse [60]. Only one study compared tanezumab with a placebo in patients with painful bone metastases. This study showed no difference in pain relief or the percentage of people who experienced pain relief between groups [61]. Because the data and evidence are limited, WHO experts have not been able to recommend or not recommend monoclonal antibodies for the treatment of CIBP [45].

#### 4.3.3. Analgesics According to the WHO Analgesic Ladder

Treatment of bone pain should take into consideration the use of analgesic drugs at any stage of disease [44]. The main aim of pain management with analgesics is to reduce pain quickly at rest and during movement, not to prevent SREs. On the basis of low-quality evidence, the WHO recommends that, for adults (including older persons) and adolescents with pain related to cancer, non-steroidal anti-inflammatory drugs (NSAIDs), paracetamol, and opioids—alone or in combination—should be used upon the initiation of pain management, depending on the clinical assessment and pain severity. The WHO recommends a three-step analgesic ladder approach based on pain intensity. At each step, adjuvant drugs addressing specific pain mechanisms should be considered [45].

In patients with mild pain intensity (i.e., score < 4 on a numerical rating scale (NRS)), non-opioid analgesics such as acetaminophen or/and NSAIDs are recommended [44,45]. NSAIDs inhibit PG synthesis via cyclooxygenases (COX) and reduce local edema and PG-induced sensitization, so clinicians consider them to be useful for CIBP because many of the symptoms are related to local inflammation and tissue injury. This opinion is based on experience rather than strong evidence, and the efficacy of NSAIDs in the treatment of CIBP in humans has not been confirmed [62]. However, in experimental studies, the inhibition of PGE2 synthesis by COX-2 inhibitors resulted in the reduced growth of bone cancer or reduced pain behavior in mice [63].

In a Cochrane Database Systematic Review, Derry et al. [64] showed that there is no high-quality evidence to support or refute the use of NSAIDs alone or in combination with opioids. Although the quality of evidence was low, moderate or severe pain was reduced to mild pain after 1 week or 2 weeks with an NSAID in 26% and 51% of patients, respectively. The main limitation of long-term treatment with NSAIDs is the risk of gastrointestinal (GI) irritation/bleeding, decreased kidney function, cardiovascular events, or bleeding disorders. Elderly patients are particularly vulnerable to all these adverse effects [64].

In patients with a contraindication to NSAIDs, acetaminophen can be considered, but it is less efficacious than NSAIDs in the treatment of bone pain. Nevertheless, no clinical studies have compared both substances directly in patients with CIBP. “Strong” opioids are the mainstay of analgesic therapy in treating moderate to severe cancer-related pain. These drugs exert their clinical effects by influencing opioid receptors, which are classified into three groups: MOR (μ), DOR (δ), and KOR (κ). Although various “strong” opioids with different pharmacological properties exist, one is not superior to another in clinical practice in terms of efficacy [44,45]. Opioids are efficacious against somatic, neuropathic, and mixed pain. They can reduce background and episodic pain in patients with CIBP [65].

Opioids are used widely in cancer patients, but studies have suggested that some of them may promote cancer progression. The main mechanisms responsible for this effect are the stimulation of angiogenesis and immunosuppression, mainly mediated by agonism at MOR [66]. Studies have suggested that morphine has the greatest immunosuppressive potential, and fentanyl has intermediate potential, whereas buprenorphine and tramadol have shown the lowest or no immunosuppressive effect [67]. Experimental studies have shown that KOR agonists are efficacious in the treatment of CIBP in mice, without changing tumor size or proliferation in cancer cell lines. These data suggest that KOR agonists could be used to manage cancer pain without the drawbacks of MOR agonists and without worsening disease progression [68].

### 4.4. Adjuvant Analgesics

#### 4.4.1. Corticosteroids

Corticosteroids are among the most commonly used adjuvant analgesics for the management of metastatic bone pain, neuropathic pain from the infiltration or compression of neural structures, as well as headaches due to increased intracranial pressure and visceral pain [69,70]. Corticosteroids exert potent anti-inflammatory and anti-swelling effects. The reduction of peritumoral edema in response to corticosteroid therapy may lead to an improvement in analgesia. Corticosteroids may also influence nociceptor activation indirectly by reducing the level of prostanoids and pro-inflammatory cytokines, or they may directly decrease the pathological electrical activity of damaged peripheral neurons, thereby decreasing the pain intensity [71]. On the basis of moderate-quality evidence, the WHO strongly recommends that adults (including older persons) and adolescents with pain related to cancer be administered adjuvant corticosteroids to achieve pain control if indicated [45].

A Cochrane Database Systematic Review conducted by Haywood et al. [72] showed that the evidence for the efficacy of corticosteroids for pain control in cancer patients is weak (although significant pain relief has been noted in some studies). The authors concluded that the side effect profile of corticosteroids (especially in the longer term) is not well established. Data on corticosteroid efficacy in patients with CIBP are sparse, so the clinical guidelines recommending the use of corticosteroids to treat CIBP are based on expert opinions rather than strong evidence [73,74]. In general, corticosteroids should be prescribed for as short a period as possible, and clinicians’ decisions should be individualized for each patient. Clinicians should take into account the location and type of pain, the presence of risk of infection, diabetes mellitus, disease stage, and concomitant therapies [45].

Corticosteroids with the least mineralocorticoid effect are preferable. Dexamethasone is the most commonly used corticosteroid because of its long half-life and lack of mineralocorticoid activity, which results in less fluid retention. The doses range from moderate (8 mg/day) to ultra-high (36–96 mg/day preceded by a bolus of 10–100 mg i.v.). Corticosteroids are usually tapered over 2 weeks [44,45].

#### 4.4.2. Anticonvulsants

Anticonvulsants are used widely and recommended for different neuropathic pain syndromes in humans [75]. Animal studies have reported the efficacy of α2δ subunit ligands in CIBP models, as well. In these models, excitability in dorsal horn processing pathways was increased, resulting in hypersensitivity. Gabapentin, acting on an α2δ subunit of a calcium channel, minimized the central sensitization and reduced pain behavior [76]. Experimental studies have suggested that pregabalin and gabapentin may be of clinical value for treating patients with CIBP. One case report that evaluated gabapentin use in patients with CIBP showed a possible positive effect [77]. Data from experimental studies and clinical observations have resulted in increased use of pregabalin or gabapentin in clinical settings, but the scientific evidence is inconsistent. One randomized controlled trial (RCT) showed a beneficial effect of pregabalin in patients with CIBP. A greater proportion of patients treated with pregabalin reported pain reduction of 30% or 50% versus the placebo, but the effect size was small [78]. However, one RCT on pregabalin in patients with CIBP showed no beneficial effect in terms of pain scores, pain interference, or QoL in comparison with the placebo. Patients treated with pregabalin reported an improvement in mood and fewer episodes of breakthrough pain [79]. One explanation for this inconsistency in clinical trials is that a neuropathic component can be identified in only approximately 17% of patients with CIBP. Identification of the mechanisms underlying a pain syndrome in a given patient (particularly the neuropathic component) could improve the efficacy of antiepileptic agents in the treatment of CIBP and the quality of pain management in cancer patients [79,80]. To date, evidence from clinical trials suggests a lack of efficacy of pregabalin and gabapentin in patients with CIBP [81,82].

## 5. Emerging Experimental Studies in Cancer-Induced Bone Pain

Experimental studies have shown that the invasion of cancer cells can lead to various physiological and morphological changes in astrocytes in the spinal cord. The enhanced activity of astrocytes affects the extracellular environment of neurons and may result in the increased release of pro-inflammatory and pronociceptive cytokines. In a sarcoma model in mice, thalidomide (an immunomodulatory and anti-inflammatory drug) reduced pain behaviors by suppressing the reactivation of astrocytes in the spinal cord and significantly decreased the expression of glial fibrillary acidic protein (GFAP) and transcriptional nuclear factor kappa B (NF-κB), which is required for the transcription of inflammatory cytokines [83].

Experimental studies have shown alterations in sphingolipid metabolism in the spinal cord in rodent models of CIBP. Following tumor implantation in mice, levels of ceramides decreased, with corresponding increases in sphingosine and sphingosine 1-phosphate (S1P), which is a potent inflammatory metabolite. S1P may play a role in neuropathic pain, as well. The S1P receptor subtype 1 (S1PR1) might act as a novel target for therapeutic intervention in CIBP because S1PR1 antagonists have been shown to attenuate cancer-induced spontaneous flinching and guarding [84].

Connexins are the key proteins in cell–cell communication and may affect CIBP at multiple levels, including nociceptive signaling and bone degradation. Carbenoxolone is a broad-acting connexin blocker. In a mouse model of CIBP, chronic systemic administration of carbenoxolone caused a significantly delayed onset and reduction of pain behavior. These results suggest that connexins are involved in CIBP and that carbenoxolone could be a novel analgesic treatment for the pain state [85].

It has been demonstrated that several breast tumor cell lines release glutamate in vitro through the cystine/glutamate antiporter system xc− (xCT). System xc− is crucial for maintaining intracellular and extracellular levels of antioxidants (e.g., glutathione, cysteine) by exchanging extracellular cystine for intracellular glutamate. System xc− is upregulated in many tumor types in response to the high concentrations of reactive oxygen species (ROS) produced as a consequence of cell growth and metabolism. Glutamate released from cancer cells may directly activate nociceptors by *N*-methyl-d-aspartate (NMDA), α-amino-3-hydroxy-5-methyl-4-isoxazolepropionic acid (AMPA), and metabotropic-type glutamate receptors on peripheral endings, resulting in pain. The experimental study of Slosky et al. confirmed that (i) tumor cell glutamate release through system xc− in vivo is a critical contributor to CIBP, (ii) ROS regulate the cystine/glutamate antiporter system xc− in tumor cells, and (iii) upregulation of system xc− increases glutamate in the bone tumor microenvironment, resulting in pain behavior in CIBP [86].

In an animal model of CIBP, repeated administration of sulfasalazine (SSZ), an inhibitor of system xc−, attenuated pain-related behavior [87]. However, in humans, SSZ has very limited oral bioavailability, inactive colonic metabolites, and a high risk of gastrointestinal side effects, which limit its use in daily clinical practice.

The study of Fazzari et al. [88] focused on the influence of capsazepine (CPZ, a competitive antagonist of the vanilloid receptor 1 TRPV-1) on system xc− in the cell culture of human breast adenocarcinoma and in an animal bone cancer model. Their results suggest that CPZ can modulate the activity of system xc− in vitro and in vivo since the systemic administration of CPZ delayed the onset and reversed CIBP-induced nociceptive behaviors in the model. This study confirms the crucial role of the cysteine/glutamate antiporter system and redox homeostasis in cancer-related pain. Potentially, direct action on cancer cells and their environment may decrease the nociception, and this approach might reduce the requirement for centrally acting analgesics in the future.

Interleukin 6 (IL6) is a pro-inflammatory cytokine that has been shown to promote pain by sensitizing nociceptors and amplifying signaling at the site of injury or disease. It plays a causal role in chronic inflammatory and immune diseases. Several studies have demonstrated that a blockade of IL6 signaling decreases responses to evoked noxious or normally non-noxious stimuli (i.e., hypersensitivity to thermal and mechanical stimulation) in a variety of preclinical models of pain. [89,90]. The study of Remeniuk et al. showed increased IL6 levels in both the local bone microenvironment and the blood serum of tumor-bearing rats. In the study, the novel molecule antagonist TB-2-081 blocked tumor-induced bone remodeling through a blockade of IL6 receptors on osteoblasts, resulting in reduced RANKL production. The blockade of IL6 signaling modified disease progression, including the prevention of tumor-induced osteoclastogenesis, bone loss, and fracture and the attenuation of pain. The results of the study indicate that, in clinical practice, the blockade of IL6 signaling by treatments such as immunotherapy may result in the prevention or delay of bone remodeling and fracture, as well as decreased pain intensity [91].

## 6. Summary

Bone metastases are common complications of cancer. They cause pain, induce pathological fractures (with all their adverse consequences), negatively affect daily functioning and QoL, and worsen the prognosis. The mechanisms leading to bone pain are complex and evolve with cancer progression. Tumor cells, bone matrix cells, as well as inflammatory cells produce mediators that activate nociceptors, damage nerve fibers, and secrete the growth factors responsible for the pathological growth and reorganization of the sensory and sympathetic nerve fibers that innervate the bones and for neuroma formation. All of these processes lead to peripheral and central sensitization. The mutual interactions between metastatic cancer cells and osteoclasts constitute a vicious circle, which leads to bone destruction, pathological fractures, and refractory pain. CIBP is multifactorial, so different strategies for different disease stages and pain mechanisms should be applied to achieve optimal pain relief and QoL. The therapeutic goal is not only pain relief but also the prevention of pain progression and SREs. Therefore, the treatment of bone pain in cancer patients should be multimodal (pharmacological and non-pharmacological), including causal anticancer and symptomatic analgesic treatment. All interventions should be individualized and directed at relieving pain, improving the QoL and functioning of patients, and, whenever possible, prolonging survival.

## Figures and Tables

**Figure 1 ijms-20-06047-f001:**
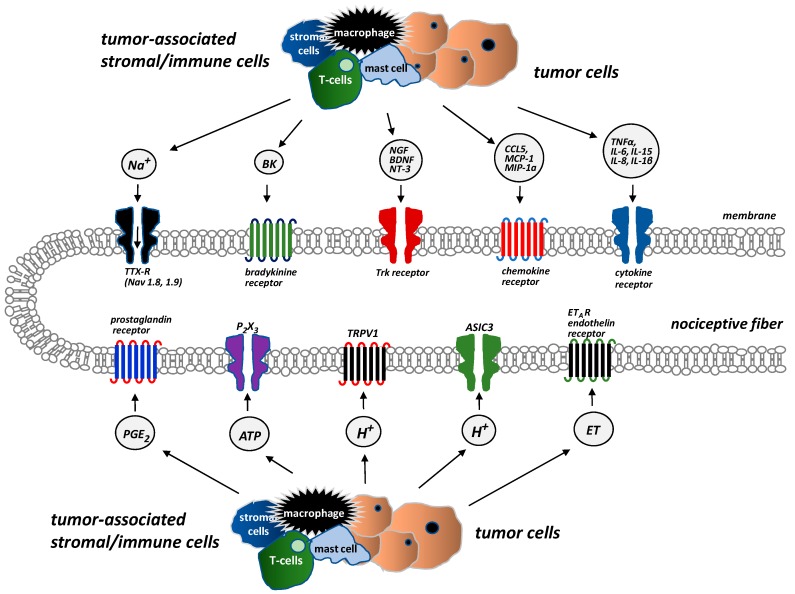
Receptors and ion channels on bone sensory nerves (According to [14]). Several types of receptors and ion channels are located on bone-innervating sensory nerves. They help to detect and transmit signals about noxious stimuli produced by tumor cells, stromal cells, and tumor-associated immune cells. These receptors (endothelin receptor ET_A_R, prostaglandin receptors, tropomyosin kinase A (TrkA) receptor, bradykinin receptor, cytokine receptors, chemokine receptors, TRPV1 (transient receptor potential channel, vanilloid subfamily member 1), acid-sensing ion channel 3 (ASIC3), purinergic receptor (P2X3)) are activated by mediators, e.g., endothelin (ET), prostaglandin E2 (PGE2), nerve growth factor (NGF), bradykinin, pro-inflammatory cytokines (tumor necrosis factor α (TNFα), interleukins (IL1β, IL6, IL8, IL15)), chemokines (CCL5, monocyte chemoattractant protein 1 (MCP-1), macrophage inflammatory protein-1a (MIP–1a)), H^+^, and extracellular adenosine triphosphate (ATP).

**Figure 2 ijms-20-06047-f002:**
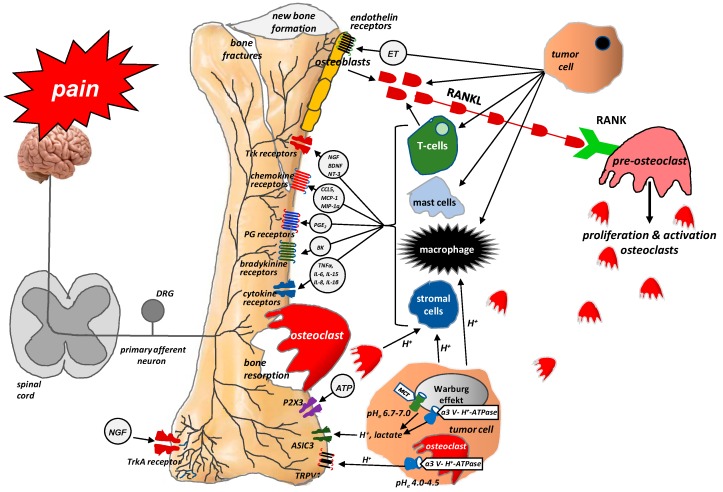
Interactions between tumor cells, osteoblasts, osteoclasts, stromal cells, and inflammatory cells at the site of bone metastasis (According to [12]). Tumor cells release endothelin (ET), which, via its appropriate receptors, interacts with osteoblasts to stimulate their proliferation. Activated osteoblasts release RANKL, which, in turn, provides a signal for the proliferation and maturation of osteoclasts and, hence, their destructive effect on bones. Osteoclasts produce ATP and acidosis-inducing H^+^, which activate appropriate receptors (P2X, TRPV1, ASICS3) located on bone-innervating neurons. Moreover, tumor cells, stromal cells, and activated immune cells secrete several mediators (endothelin, prostaglandin, NGF, bradykinin, pro-inflammatory cytokines, chemokines, H^+^, and ATP), which activate appropriate receptors (endothelin receptor, prostaglandin receptor, TrkA receptor, bradykinin receptor, cytokine receptors, chemokine receptors, TRPV1, ASIC3, P2X3) located on bone-innervating nerve fiber endings. These receptors help to detect and transmit signals about noxious stimuli to the spinal cord and then to the cerebral cortex, where perception takes place.

**Figure 3 ijms-20-06047-f003:**
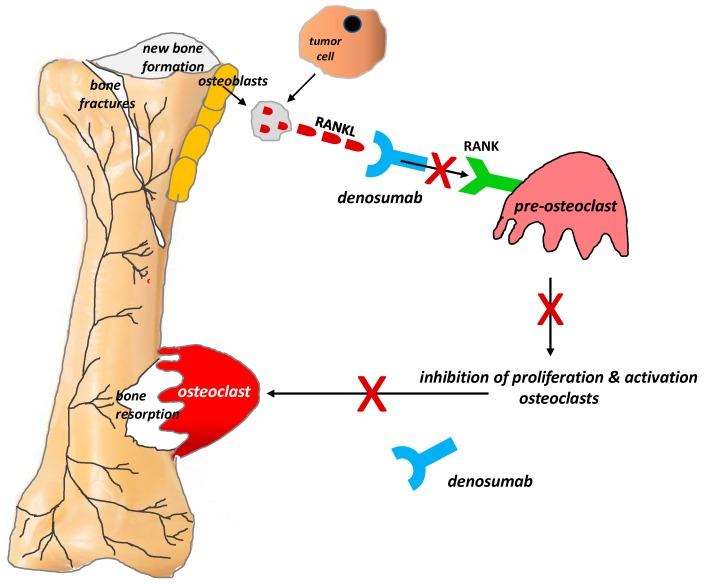
Role of the RANKL/RANK system and the mechanism of action of denosumab in bone pain due to cancer (According to [12]). RANKL is a protein secreted by tumor cells and the osteoblastic cell line (i.e., by mature osteoblasts and their precursors). RANKL binds to RANK receptors located on the osteoclast surface, which triggers the proliferation and maturation of osteoclasts, thereby initiating their damaging effect on bones. Denosumab is characterized by a high affinity and specificity for RANKL. It prevents the activation of RANK receptors on the surface of osteoclasts and their precursors, thereby inhibiting the formation, proliferation, and survival of osteoclasts, which reduces osteoclastic bone resorption. **X** = stop.

**Figure 4 ijms-20-06047-f004:**
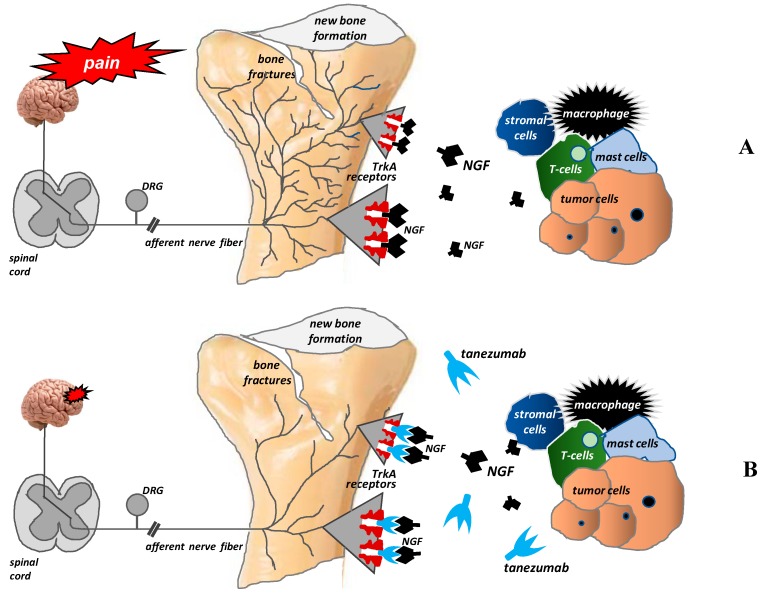
Pathological nerve sprouting in bone metastases and the mechanisms of action of tanezumab (according to [42]). After binding to the TrkA receptor, nerve growth factor (NGF) activates the pathological formation and growth of a network of new nerve fibers characterized by a unique morphology, organization, and density, which exceeds the density of nerve fibers in normal bone by up to 10–70 times. These new pathological nerve fibers are formed within the periosteum, mineralized bone tissue, and bone marrow (**A**). Tanezumab, a humanized anti-NGF monoclonal antibody, prevents NGF from binding to the TrkA receptor and consequently inhibits the pathological sprouting and formation of new pathological nerve fiber networks (**B**).

**Figure 5 ijms-20-06047-f005:**
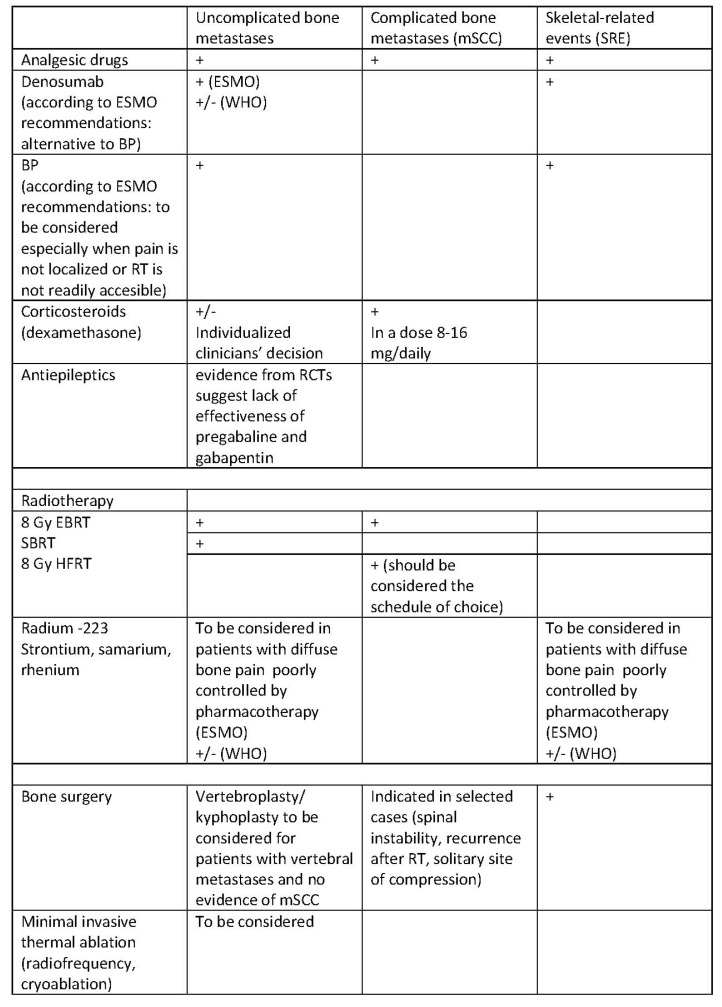
Treatment of pain due to bone metastases according to ESMO and WHO recommendations [44,45]. mSCC, metastatic spinal cord compression; SRE, skeletal-related events; BP, bisphosphonates; EBRT, external beam radiotherapy; SBRT, stereotactic body radiotherapy; HFRT, hypofractionated radiotherapy; RT, radiotherapy; (+) recommended; (+/−) experts have not made a recommendation for or against the given method of treatment.

**Figure 6 ijms-20-06047-f006:**
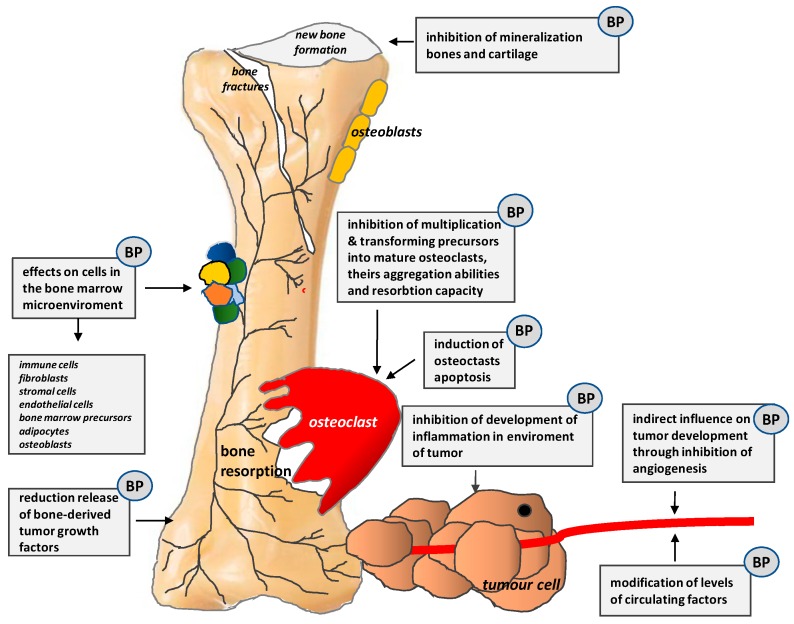
Mechanisms of action of bisphosphonates (BPs) (According to [13]).

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
