# Peer review of "Bone Pain in Cancer Patients: Mechanisms and Current Treatment"

_ijms, 2019, doi:10.3390/ijms20236047_

Round 1

Reviewer 1 Report

I have peer reviewed the manuscript titled “Bone pain in cancer patients – mechanisms and current treatment” submitted to the IJMS. The authors have comprehensively reviewed mechanism and various management options for cancer associated bone pain. The manuscript is comprehensive, very detailed and will be an interesting contribution for the journals’ audience. The manuscript needs some major improvements; there are a few suggestions that authors may consider to improve it further:

There are a number of grammatical errors and unclear statements; there are a number of statements in the abstract requiring rephrasing. What is the aim or objective of this article; and how will it benefit the scientific community? should be clearly described in the abstract as well as introduction.

The abstract is unstructured; however; to the point.

The captains for figures 2 and 3 should more details explain the components of the image in context. For example, what is meaning of red “x” or other coloured markers. Adding such information will be good for the reader.

Authors should be consistent for using terms: the manuscript used the term “tumour” on mostly places; however mixed up with “tumor” on certain places such as lines 158, 171, and figure 6. Please correct.

Regarding the section 4 (4. Treatment of cancer–induced bone pain)

Authors did not mention about the gene therapy that may have applications for the management of cancer associated bone pain. For example, see the following article that can be included;

Gene therapy: A paradigm shift in dentistry. Genes. 2016 Nov;7(11):98.

Regarding the section; “Emerging experimental studies in cancer–induced bone pain”.  Authors included only three resent studies in their discussion; however a lot of research is on way on this area. Therefore, adding more studies and expanding this section will improve this manuscript.

Figures have been presented nicely; however, it is not clear either authors have reproduced these images from the mentioned source or their own;

If reproduced; only citation may not be enough and may require permission/copyright.

Author Response

Response to Reviewer 1 Comments

I have peer reviewed the manuscript titled “Bone pain in cancer patients – mechanisms and current treatment” submitted to the IJMS. The authors have comprehensively reviewed mechanism and various management options for cancer associated bone pain. The manuscript is comprehensive, very detailed and will be an interesting contribution for the journals’ audience. The manuscript needs some major improvements; there are a few suggestions that authors may consider to improve it further:

There are a number of grammatical errors and unclear statements; there are a number of statements in the abstract requiring rephrasing. What is the aim or objective of this article; and how will it benefit the scientific community? should be clearly described in the abstract as well as introduction.The abstract is unstructured; however; to the point.

As suggested by the Reviewer, we modified the abstract, shortened it, and included the aim the paper as well as its clinical relevance. We also corrected the grammar and language of the manuscript. Please, find attached certificate of English editing 

The captains for figures 2 and 3 should more details explain the components of the image in context. For example, what is meaning of red “x” or other coloured markers. Adding such information will be good for the reader.

The legend below each figure explains the mechanisms illustrated in it (they are highlighted in yellow). As suggested by the Reviewer, we have also included an explanation of red "x."

Authors should be consistent for using terms: the manuscript used the term “tumour” on mostly places; however mixed up with “tumor” on certain places such as lines 158, 171, and figure 6. Please correct.

As suggested by the Reviewer, we have corrected.

Regarding the section 4 (4. Treatment of cancer–induced bone pain)

Authors did not mention about the gene therapy that may have applications for the management of cancer associated bone pain. For example, see the following article that can be included;

Gene therapy: A paradigm shift in dentistry. Genes. 2016 Nov;7(11):98.

The authors of manuscript focused mainly on studies and recommendations regarding clinical practice in patients with cancer associated bone pain. Neither ESMO nor WHO recommend gene therapy as a standard treatment of pain related to bone metastases, therefore the authors did not include this method in the manuscript. The available papers describe gene therapy for cancer in itself rather than for pain management, so the authors decided to not add this method to the manuscript.

Regarding the section; “Emerging experimental studies in cancer–induced bone pain”.  Authors included only three resent studies in their discussion; however a lot of research is on way on this area. Therefore, adding more studies and expanding this section will improve this manuscript. 

The authors are aware that manuscript is long, so they decided to not describe all available experimental trials on CIBP, because these methods are still not regarded as current clinical treatment modalities. They probably may be introduced into the clinical practice in the future, so the idea of authors was to show most current developments and experiments in the animal models of bone pain.  However, according to Reviewer’s suggestion we added information regarding other experimental studies (highlighted in yellow).

Figures have been presented nicely; however, it is not clear either authors have reproduced these images from the mentioned source or their own;

If reproduced; only citation may not be enough and may require permission/copyright.

According to the Reviewer’s suggestion we have redrawn the Figure 5.

All Figures (1-6) have been prepared by the authors of the manuscript, based on sources which we quote in brackets.

Reviewer 2 Report

In this review manuscript the authors describe the mechanisms of bone pain in cancer patients and the currently available treatment strategies.  Moreover the authors explain the complex pathophysiology of metastatic induced bone pain and provide information on clinical characteristics of bone pain in cancer patients.

General comments

Many cancers have a high propensity to metastasize to the skeleton. As soon as tumor cells have spread to bone, cancers are usually considered as incurable. Moreover, bone metastases have devastating consequences resulting not only in pathological fractures but also in life-threatening hypercalcemia, nerve compression syndromes and severe pain. Although therapeutic strategies have consistently improved over the past decades, the overall prognosis of patients suffering from metastatic bone disease remains poor. For this reason, enhanced knowledge on different treatment strategies including pain management is of high clinical relevance.

Specific comments

This manuscript is written well and it comprises all important aspects on pain management of patients with metastatic bone disease. However the manuscript requires some changes if it is to be a resource to the readership of this journal.

In general, the present manuscript is too long and it would benefit from a re-organization and more efficient and concise writing. For instance, some parts are repetitive which is unnecessary as they do not provide crucial information. (e.g. lines 286, 297 and 407)

Although the topic is of high significance, considerable novelty is missing. I suggest that the authors focus on giving an up to date summary on currently available treatment strategies. For clinicians, it would be desirable to have an overview – maybe in a tabulated form – clearly stating what kind of treatment options are indicated at certain stages of the disease.

The authors clearly describe the metastatic process and its different stages, however, an important aspect is missing. It should be mentioned that, although the precise mechanisms are unclear, it appears that upon arrival in bone some cancer cells enter into a state of dormancy, which apparently can last for many months if not years.

Collectively, the manuscript is written well and it illustrates an interesting and important topic. As such, the manuscript is likely to be of interest to the vision of clinicians, researchers and scientists for this journal.

Author Response

In this review manuscript the authors describe the mechanisms of bone pain in cancer patients and the currently available treatment strategies.  Moreover the authors explain the complex pathophysiology of metastatic induced bone pain and provide information on clinical characteristics of bone pain in cancer patients.

General comments

Many cancers have a high propensity to metastasize to the skeleton. As soon as tumor cells have spread to bone, cancers are usually considered as incurable. Moreover, bone metastases have devastating consequences resulting not only in pathological fractures but also in life-threatening hypercalcemia, nerve compression syndromes and severe pain. Although therapeutic strategies have consistently improved over the past decades, the overall prognosis of patients suffering from metastatic bone disease remains poor. For this reason, enhanced knowledge on different treatment strategies including pain management is of high clinical relevance.

Specific comments

This manuscript is written well and it comprises all important aspects on pain management of patients with metastatic bone disease. However the manuscript requires some changes if it is to be a resource to the readership of this journal.

In general, the present manuscript is too long and it would benefit from a re-organization and more efficient and concise writing. For instance, some parts are repetitive which is unnecessary as they do not provide crucial information. (e.g. lines 286, 297 and 407)

A legend under each figure explains the mechanisms shown in it. It should not be treated as a repetition of parts of the text, but as an explanation of the contents of the figure, which is useful for the reader and makes the manuscript clearer. For example, lines 162−172 are the legend for Figure 1, lines 192−204 for Figure 2, lines 240−247 for Figure 3, lines 290−297 for Figure 4, and lines 318−322 for Figure 5. Legends to the figures are highlighted in yellow.

Although the topic is of high significance, considerable novelty is missing. I suggest that the authors focus on giving an up to date summary on currently available treatment strategies. For clinicians, it would be desirable to have an overview – maybe in a tabulated form – clearly stating what kind of treatment options are indicated at certain stages of the disease.

The authors of manuscript have re-arranged the Figure 5 according to the Reviewer’s suggestion.

The authors clearly describe the metastatic process and its different stages, however, an important aspect is missing. It should be mentioned that, although the precise mechanisms are unclear, it appears that upon arrival in bone some cancer cells enter into a state of dormancy, which apparently can last for many months if not years.

As suggested by the Reviewer, we have added this interesting piece of information to the paper (highlighted in yellow).

Collectively, the manuscript is written well and it illustrates an interesting and important topic. As such, the manuscript is likely to be of interest to the vision of clinicians, researchers and scientists for this journal.

Round 2

Reviewer 1 Report

Many thanks for the revision and incorporating all suggested changes to the manuscript